# The Anti-Cancer Activity of Pentamidine and Its Derivatives (WLC-4059) Is through Blocking the Interaction between S100A1 and RAGE V Domain

**DOI:** 10.3390/biom13010081

**Published:** 2022-12-30

**Authors:** Nuzhat Parveen, Wei-Jung Chiu, Li-Ching Shen, Ruey-Hwang Chou, Chung-Ming Sun, Chin Yu

**Affiliations:** 1Chemistry Department, National Tsing Hua University, Hsinchu 300, Taiwan; 2Department of Applied Chemistry, National Yang Ming Chiao Tung University, Hsinchu 300, Taiwan; 3Graduate Institute of Biomedical Sciences, China Medical University, Taichung 40402, Taiwan; 4The Ph.D. Program of Biotechnology and Biomedical Industry, China Medical University, Taichung 40402, Taiwan; 5Center for Molecular Medicine, China Medical University Hospital, Taichung 40402, Taiwan; 6Department of Medical Laboratory Science and Biotechnology, Asia University, Taichung 41354, Taiwan; 7Department of Medicinal and Applied Chemistry, Kaohsiung Medical University, Kaohsiung 807, Taiwan

**Keywords:** protein–ligand interaction, small compound WLC-4059, cell proliferation, docking, cytotoxicity

## Abstract

The S100A1 protein in humans is a calcium-binding protein. Upon Ca^2+^ binding to S100A1 EF-hand motifs, the conformation of S100A1 changes and promotes interactions with target proteins. RAGE consists of three domains: the cytoplasmic, transmembrane, and extracellular domains. The extracellular domain consists of C1, C2, and V domains. V domains are the primary receptors for the S100 protein. It was reported several years ago that S100A1 and RAGE V domains interact in a pathway involving S100A1-RAGE signaling, whereby S100A1 binds to the V domain, resulting in RAGE dimerization. The autophosphorylation of the cytoplasmic domain initiates a signaling cascade that regulates cell proliferation, cell growth, and tumor formation. In this study, we used pentamidine and a newly synthesized pentamidine analog (WLC-4059) to inhibit the S100A1-RAGE V interaction. ^1^H-^15^N HSQC NMR titration was carried out to characterize the interaction between mS100A1 (mutant S100A1, C86S) and pentamidine analogs. We found that pentamidine analogs interact with S100A1 via ^1^H-^15^N HSQC NMR spectroscopy. Based on the results, we utilized the HADDOCK program to generate structures of the mS100A1–WLC-4059 binary complex. Interestingly, the binary complex overlapped with the complex crystal structure of the mS100A1–RAGE-V domain, proving that WLC-4059 blocks interaction sites between S100A1 and RAGE-V. A WST-1 cell proliferation assay also supported these results. We conclude that pentamidine analogs could potentially enhance therapeutic approaches against cancers.

## 1. Introduction

The family of human Sl00 proteins contains more than 19 members, which are located as a cluster on human chromosome lq21 [1]. S100-family proteins are small acidic dimeric proteins with a molecular weight of around 11 kDa and contain 2 EF-hand motifs for Ca^2+^ binding [2,3]. Each member exhibits a unique spatial and temporal expression pattern. The S100A1 protein is an Sl00 protein that is homodimeric or heterodimeric with S100Al/P, S100A1/B, and S100A1/A4 [4]. In addition to being found in the brain, the S100Al protein is highly expressed in the heart, thyroid gland, skeletal muscle, skin, kidneys, salivary glands, and breasts [5]. It has also been associated with cardiomyopathy, neurodegenerative disorders, endometrial cancers, renal cancers, and other diseases [6,7].

Helixes 3 and 4 in S100A1 reorient upon calcium binding, which exposes a large hydrophobic pocket between them [8,9]. This process is involved in interactions with most calcium-dependent target proteins. In previous studies, S100A1 was found to interact with other proteins, such as RyR1, RyR2, ATP2A2, TRPM3, RAGE, and others [10]. These proteins induce activities or conformational changes in S100Al that promote specific physiological functions [2,11,12]. Sl00Al gene therapy has been developed and implemented in its first human clinical trials in recent years.

RAGE is in the immunoglobulin superfamily and has a role in signal transduction at cell surface receptors [13,14,15]. RAGE is 35 kDa (382 amino acids) and has five domains: constant domains (C1 and C2) and one variable domain (V) at the N-terminus, a cytoplasmic domain for signal transmission at the C-terminus, and a transmembrane domain for membrane anchoring [16]. As the major binding domain of RAGE, the V domain binds to almost every S100-family ligand, along with advanced glycation end-products (AGEs), amphoterin, high-mobility group protein 1 (HMGB1), amyloid-β protein, etc. [17]. Some proteins, such as S100A12 and S100B, can bind not only to the V domain but also to the C1 domain of RAGE [18].

Previous studies showed that RAGE can combine to form an oligomer [19,20]. The ligand binds to RAGE’s tail-like cytoplasmic domains, which leads to autophosphorylation and promotes intracellular signaling [21]. Various diseases are triggered by these factors, including diabetes, neurodegeneration, chronic vascular inflammation, and cancer [14,22]. RAGE can form ligand-specific homodimers that are involved in amplifying signal transduction and transcriptional activation via the mitogen-activated protein kinase (MAPK) and Cdc42/Rac pathways, as well as the transcription factors NF-kappaB and AP-1 [23]. Cell proliferation, migration, and tumor growth are also induced by signal transduction. Therefore, it is necessary to improve or develop treatments for RAGE-dependent diseases based on the knowledge of how RAGE interacts with its targets.

Pentamidine (4,4′-[pentane-1,5-diylbis(oxy)] dibenzenecarboximidamide; see Figure 1) is an orphan drug used for acute protozoa-dependent illnesses, such as preventing protozoal infections in some patients with constantly depressed immune systems who have undergone organ transplantation [24].

Pentamidine is also used for the treatment of *Pneumocystis carinii* pneumonia (PCP), leishmaniasis, yeast infections caused by *Candida albicans*, and *Acanthamoeba* infections in immunocompromised patients and as a prophylactic antibiotic for children undergoing treatment for leukemia [25].

In previous research, pentamidine was also indicated as an anti-tumor drug [26,27]. Furthermore, we found that pentamidine and its derivatives can be effective inhibitors by blocking the interface between S100A1 and the RAGE V domain. This was identified by using NMR ^1^H-^15^N HSQC titration, the binding constant (K_d_) evaluated by fluorescence, and the heterodimeric complex structure obtained from calculations using the HADDOCK program. The structure of the newly synthesized compound (WLC-4059, a pentamidine derivative) is shown in Figure 1.

## 2. Methods

### 2.1. S100A1 Expression and Purification

The structure and sequence of the human calcium-S100A1 homodimer were reported several years ago using NMR [28]. The NMR spectrum accuracy is affected by the presence of a free cysteine at residue 86 of S100A1 [29]. We purchased a mutant cDNA clone of mutant S100A1 (mS100A1), where Cys 86 is mutated to serine. Therefore, the NMR detection of mS100A1 ^15^N-^1^H HSQC could be executed without DTT.

The cross-peaks in the ^1^H-^15^N HSQC spectrum of human calcium-S100A1 and calcium-mS100A1 had high similarity and overlapped well. This suggested that not only would the three-dimensional structures of S100Al be similar between the wild type and mutant type, but so would their chemical and physical properties, such as hydrophobicity, according to predictions using the ExPASy ProtParam tool. The recombinant cDNA of the protein sequence (1–94) of mS100A1 was inserted at Ndel and Xhol restriction sites and then cloned into the pET-20b expression vector. This was transferred and expressed in BL21(DE3) from Novagen.

^15^N-labeled mS100A1 was prepared by culturing *E. coli* containing the mS100A1 gene in M9 medium including ^15^NH_4_Cl at 310 K. Culturing was carried out until the optical density (O.D.) was 0.8–1.0 at 600 nm. *E. coli* was then induced by adding 1 mM IPTG in incubators at 310 K, followed by shaking at 200 rpm for 10 h. To collect the mS100A1 protein, mS100A1-rich *E. coli* was lysed by the French press method in resuspension buffer containing 20 mM Tris-HCl and 1 mM ethylenediaminetetraacetic acid (EDTA) at pH 7.5, which is similar to the S100A1 resuspension buffer. Most proteins would be in the supernatant fraction after centrifuging at 12,500 rpm for 1 h at 277 K.

The obtained supernatant contained mS100A1 after centrifugation and was purified on a Hi-Prep Phenyl FF 16/10 Q-Sepharose column on the AKTA system. The goal was to elute the fraction containing mS100A1 with a gradient of 0–1.0 M sodium chloride (NaCl) in a fast protein liquid chromatography system (GE Healthcare). Following the exchange of buffer to 20 mM Tris-HCl, 100 mM NaCl, and 10 mM calcium chloride (CaCl_2_) at pH 7.5, the fraction containing mS100A1 was purified on a (HIC) column using the AKTA FPLC system [30,31]. A large amount of pure mS100A1 protein (>95%) was obtained by gradient elution with a buffer containing 20 mM Tris-HCl and 20 mM EDTA at pH 7.5. Finally, the mS100A1 protein was confirmed using high-performance liquid chromatography (HPLC), sodium dodecyl sulfate polyacrylamide gel electrophoresis (SDS-PAGE), and electrospray ionization (ESI) mass spectrometry (Appendix A).

### 2.2. 2D NMR Experiments

Two-dimensional NMR was executed by adding pentamidine and its newly synthesized derivative (WLC-4059) into uniformly ^15^N-labeled mS100A1 solutions at molar ratios of 1:0, 1: 0.5, 1:1, and 1:2. VnmrJ 2.3 was used to process all NMR data, and NMRFAM-SPARKY 1.4 powered by Sparky 3.13 was used for analysis [32,33,34]. The perturbations and intensity changes of chemical shifts were identified by superimposing HSQC spectra for comparison.

### 2.3. Biomolecular Docking (HADDOCK)

We applied (HADDOCK 2.2) software to generate the structural models of S100A1–pentamidine and the pentamidine derivative. The NMR solution structural coordinates of the calcium-bound S100A1 homodimer (PDB ID: 2LP3) were obtained from the PDB [35,36,37], and those of pentamidine were obtained from DrugBank (DrugBank ID: DB00738). The PDB structures for the pentamidine derivative were generated from PyMOL. The perturbations and intensity decreases in the cross-peaks of HSQC titration were analyzed and defined as ambiguous interaction restraints on interfaces.

The active and passive residues based on the accessible surface area of either the side chain or backbone were predicted by HADDOCK 2.2 software [38,39]. One thousand structures from HADDOCK were analyzed using an explicit solvent (water) with the lowest energy, and then the side-chain contacts were refined and optimized. mS100A1–pentamidine/WLC-4059 docking-complex structures were generated. Then, all results of the structures were demonstrated in the PyMOL program [40,41].

### 2.4. WST-1 Cell Proliferation Assay

Cell proliferation assay by using the WST-1 reagent were conducted to support our NMR experiments. For pre-experiment preparation, cells were cultured for one or two days until they reached the logarithmic phase. They were then trypsinized and seeded in 96-well plates at a density of 1 × 10^4^ cells per well. Subsequently, cells were incubated for a period of 24 h in a serum-free medium that contained 0.1% BSA as a serum substitute.

Next, the serum-starved cells were treated for another 48 h with or without the indicated concentrations of recombinant mS100A1 proteins, pentamidine, and the pentamidine derivative. The WST-1 cell proliferation reagent (Roche, Dubai, United Arab Emirates) was added to each well at one-tenth volume prior to harvesting, and the cells were incubated at 37 °C for another 4 h. A shaker was used to gently agitate the medium in the cell culture plate for 10 min to mix it. The absorbance of the sample was measured at 450 nm by using a synergy 2 microplate reader. The relative cell numbers were calculated by comparing the absorbance from the control treatment.

## 3. Results and Discussion

### 3.1. Synthesis of Pentamidine Analog (WLC-4059)

The synthesis of the compound WLC-4059 is described in the following (Figure 1).

The pentamidine derivative WLC-4059 was synthesized by treating 5 4-hydroxybenzonitrile **1** with 1,2-bis(bromomethyl) benzene **2** in the presence of K_2_CO_3_ in refluxing acetone for 16 h. Next, **3** was stirred in lithium bis(trimethylsilyl) amide solution (LiHMDS, 1.0 M in THF, 20 mL) at room temperature overnight. After the reaction was completed, the reaction mixture was quenched with a 2.0 M HCl solution in an ice bath, and in the next phase, the mixture was stirred at room temperature for another two hours. A suspension was obtained by removing the solvent under reduced pressure. The suspension was filtered to obtain WLC-4059 as a white solid in 74% yield. The identity of WLC-4059 was confirmed by ^1^H NMR and HRMS (copies of the characterization data are provided in the Appendix A).

### 3.2. Chemical Synthesis of 4,4′-((1,2-Phenylenebis(methylene))bis(oxy))dibenzonitrile (3)

To the solution of 4-hydroxybenzonitrile **1** (1.0 g, 8.4 mmol, 2.0 eq.) in acetone (50 mL) was added 1,2-bis(bromomethyl)benzene **2** (1.1 g, 4.2 mmol, 1.0 eq.) and K_2_CO_3_ (4.0 g, 21.0 mmol, 5 eq.). The reaction mixture was refluxed for 16 h. A white solid was obtained from the crude by concentrating it in vacuo after the reaction was completed by filtering it through celite. The purification of the crude product was accomplished using column chromatography (eluted with EtOAc–Hexane = 1:1) to obtain the desired product as a white solid (1.35 g, 94.5% yield). 1 H NMR (400 MHz, CDCl_3_) δ = δ 7.57 (d, J = 9.0 Hz, 4H), 7.49 (dd, J = 5.4, 3.4 Hz, 2H), 7.43–7.39 (m, 1H), 6.98 (d, J = 9.0 Hz, 4H), 5.19 (s, 4H).

### 3.3. Chemical Synthesis of 4,4′-((1,2-Phenylenebis(methylene))bis(oxy))dibenzimidamide (WLC-4059)

A solution of **3** (0.500 g, 1.470 mmol, 1.0 eq.) in lithium bis(trimethylsilyl)amide solution (LiHMDS, 1.0 M in THF, 20 mL) was stirred at room temperature for 2 h. In an ice bath, the reaction mixture was quenched with a 2 M HCl solution and then stirred for another two hours at room temperature. Under reduced pressure, the solvent was removed to obtain a suspension. A white solid was obtained from the suspension by filtering (0.4 g, 74% yield). ^1^H NMR (600 MHz, DMSO-d_6_) δ = 7.96 (d, J = 8.0 Hz, 4H), 7.54–7.49 (d, J = 2.4 Hz, 2H), 7.36–7.33 (d, J = 2.3 Hz, 2H), 7.18–7.14 (d, J = 7.9 Hz, 4H), 5.33 (s, 4H), ^13^C NMR (400 MHz, DMSO-d_6_) δ = 165.2, 162.9, 135.1, 130.8, 129.3, 120.1, 116.4, 115.5, 68.0, 40.3; HRMS (ESI) calcd for C_22_H_22_N_4_O_2_M+H]^+^: 375.1821; found 375.1824.

### 3.4. Mapping mS100A1 and RAGE V Domain Binding Interface

HSQC NMR experiments are commonly used to identify protein–ligand or protein–protein binding interfaces. The residues of mS100A1 and its ligands (pentamidine and the newly synthesized pentamidine derivative) at their interfaces could be identified by observing the resonances in NMR ^1^H-^15^N HSQC spectra of free mS100A1 compared with mS100A1 in complex with pentamidine and the pentamidine derivative. Overlapped HSQC spectra of free ^15^N-labeled mS100A1 with pentamidine are shown in Figure 2A-1, and the residues showing chemical shift changes after titration are shown in Figure 2A-2 in an illustration of S100A1 (in stick form, colored in red). Additionally, the overlapped HSQC spectra of free ^15^N-labeled mS100A1 with the pentamidine derivative (WLC-4059) are shown in Figure 2B-1, and the residues showing chemical shift changes after titration with the pentamidine derivative (WLC-4059) are shown in Figure 2B-2 in an illustration of S100A1 (stick form, colored in red).

Parts of NMR signals were decreased after adding free pentamidine and WLC-4059 to ^15^N-labeled mS100A1 in the complex. The NMR signals of mS100A1 in complex with pentamidine and WLC-4059 residues at the interface were decreased significantly compared to those of free mS100A1. This phenomenon was due to the affected nuclei at the binding interface of the proteins between free mS100A1 and mS100A1 in complex with pentamidine and WLC-4059. In the NMR spectrum, cross-peaks decreased as the surrounding nuclei at the interface were enclosed by the other protein.

The overlapping of the ^1^H-^15^N HSQC spectra of free mS100A1 and the complex of mS100A1 with pentamidine and WLC-4059 was determined to identify mS100A1 residues. A map of the structure of mS100A1 was constructed using the residues of the HSQC cross-peaks with severe intensity decreases. Most residues from the pentamidine titration were at the position between helix 3, the linker region, and helix 4 of mS100A1, such as L28, S29, L37, G43, F44, L61, F71, V76, L77, L81, T82, and W90, as shown in Figure 2A-1. A representation of the perturbed residues (in stick form, colored in red) is shown in Figure 2A-2 in a diagram of S100A1. The perturbed residues after titration with WLC-4059 were G43, F44, A47, V76, T82, C85, and N87, as shown in Figure 2B-1, and a representation of the perturbed residues (in stick form, colored in red) is shown in Figure 2B-2 in a cartoon form of S100A1.

### 3.5. Complex Formation of S100A1 with the Small Molecule Pentamidine and the Pentamidine Derivative (WLC-4059)

The binding of Ca^2+^ S100A1 has an important role in multiple cancers when interacting with the RAGE V domain when signals cascade in cancer cells. We looked for a small molecule to act as an inhibitor that efficiently blocks the interaction between S100A1 and the RAGE V domain to prevent cancers. The small molecules pentamidine and the pentamidine derivative WLC-4059 (Figure 1) were used to bind to S100A1. We used HSQC titration experiments to identify whether the drugs interacted with S100A1 and the interfaces by observing changes in cross-peaks. The chemical shift perturbation was calculated and compared, and the ambiguous interaction restraints (AIRs) were identified.

HADDOCK 2.2 was used to find complexes for S100A1, pentamidine, and the pentamidine derivative [38,39,42]. The residues showing changes after the titration of S100A1 with pentamidine/the pentamidine derivative WLC-4059 at a ratio of 1:1 were used as input data for the complex formation. The PDB structure for S100A1 was derived from the PDB under PDB ID 2lp3, and the PDB structure for the pentamidine derivative was generated from PyMOL.

A total of 2000 rigid-body docking solutions were obtained from rigid-body energy minimizations, and for simulated annealing, 200 structures were selected. Finally, refinement in explicit water was performed in Cartesian space using simulated annealing. The best complexes were chosen based on the lowest energies and the HADDOCK score based on the root-mean-square deviation as shown in the (Figure 3A-1,A-2,B-1,B-2). PyMOL 2.4.0 software was used for the visualization of the HADDOCK complexes [43,44].

### 3.6. K_d_ Calculation Using Fluorescence

For the calculation of the dissociation constant (K_d_), we used fluorescence spectroscopy (F-2500 Hitachi fluorescence spectrophotometer). The experimental conditions for fluorescence were the same for all experiments. There is a tryptophan residue in S100A1 that is capable of being excited and emitting fluorescence. Excitation was carried out at 295 nm. A wavelength of 351 nm defines the maximum emission band of S100A1. The wavelengths of emission were detected in the range of 305–404 nm [8].

S100A1 at a concentration of 3.50 µM was treated with pentamidine and its derivative in increments of 2 µM in each addition. The pentamidine derivative WLC-4059 inhibited the mitogenic activity of S100A1 through the same mechanism as pentamidine. The Stern–Volmer equation was used to plot a nonlinear regression curve between activity and the drug concentration (Figure 4) [45].
(1)$$\log\;\frac{\left(F_{0}-F\right)}{F}\;\log\;K+n\;\log\;[Q]$$

*F*_0_ = intensity without compounds;

*F* = intensity with the compounds at the concentration *F* [Q].

### 3.7. Functional Assay

We used an assay experiment to support the biological functions on cell proliferation. To determine the downstream effects mediated by the human S100A1 protein, a proliferation assay can be used. The disruption of the protein–protein interaction, however, is a critical factor that affects the assay. In colorectal cancer tissues, RAGE expression has been found to be high, and its expression has been associated with a higher density of microvessels in the tumor tissue. The SW480 cell line is a colorectal cancer cell line that exhibits a high expression of RAGE. Specific knockdown of RAGE-pathway by shRNAs inhibits the invasion of colorectal cancer cells and suppresses angiogenesis [46], indicating a critical role for the RAGE pathway in SW480 cells. Our functional assay was conducted using SW480 cells treated with S100A1 for 48 h. An assay that measures the proliferation of viable cells was conducted using WST-1 reagent.

SW-480 cells that had been serum-starved for 24 h were grown under S100A1 protein treatment (100 nM) (Figure 5, Lane 1). Compared to the serum-free control group, a 1.44-fold increase was observed in the viable cell count when the S100A1 protein concentration was increased to 100 nM (Figure 5, Lane 2). After the addition of 1 µM of the RAGE V domain to SW480 cells (Figure 5, Lane 3), the proliferative activity of the SW480 cells was decreased by 1.21-fold. When pentamidine was added to the experiment, cell growth was significantly lower than that with the RAGE V domain (Figure 5, Lane 5). In addition, cell growth was decreased the most when the pentamidine derivative (WLC-4059) was applied (Figure 5, Lane 4). The results showed that disruption of the interaction between S100A1 and RAGE by RAGE V domain, pentamidine, and pentamidine derivative (WLC-4059) significantly decreased S100A1-induced cell proliferation compared to that treated with S100A1 alone.

This study has been able to demonstrate that in vitro, the S100A1 protein interacts directly with the RAGE V domain. Furthermore, when SW480 cells were treated with the extracellular S100A1 protein, the ability of the cells to proliferate was significantly increased in a dose-dependent manner. Based on the findings of these studies, it has been concluded that S100A1 induces cell proliferation through the RAGE pathway. There is a high expression of RAGE in SW480 cells, which is why SW480 cells behave this way.

In addition to S100A1, other signaling pathways may also have a role to play in regulating cell proliferation in SW480 cells as a consequence of S100A1. Using the S100A1 protein as an example, we demonstrated that the protein interacts directly with pentamidine and the pentamidine derivative in vitro. We also demonstrated that the addition of pentamidine and the pentamidine derivative to SW480 cells significantly decreased S100A1-induced cell proliferation, indicating that the complex that they formed might have lower activity towards RAGE than S100A1 only. The results of this study suggest that pentamidine and its derivatives can influence S100A1’s binding to the V domains of RAGE on the cells, which leads to the inhibition of S100A1’s ability to induce cell proliferation.

### 3.8. WST-1 Assay for Cytotoxicity Analysis

DMEM/F12 medium containing 10% fetal bovine serum (FBS) was used to culture SW480 cells at 37 °C in 5% CO_2_. On the day before the experiments were carried out, the cells were trypsinized and seeded in a 96-well plate at a density of 5000 cells per well in normal culture medium. Cells were then treated with increased concentrations (0, 1, 10, 20, and 50 µM) of pentamidine and pentamidine derivative (WLC-4059) for 48 h.

Subsequently, WST-1 reagent was added at one-tenth of the total volume, followed by incubation at 37 °C for another 4 h. The relative viable cells were determined by comparing the absorbance at 450 nm (OD_450nm_) measured by using a synergy II microplate reader (BioTek Instruments, Inc., Winooski, VT, USA) of each sample from the untreated control group [47]. As shown in Figure 6 here was a significant reduction in the cytotoxicity of the pentamidine derivative (WLC-4059) comparing to pentamidine, particularly when the dose was higher than 10 µM.

Accumulated evidence has indicated that the activation of the Receptor for Advanced Glycation End-products (RAGE) facilitates the progression of colorectal cancer [48]. We compared the expression level of RAGE in tumorous and normal tissues from patients with colorectal cancer from the GEPIA database and found that the expression of RAGE is significantly higher in tumorous tissues, while its level is lower in normal tissues (Appendix A). In addition, high expression levels of RAGE in patients resulted in significantly worse overall survival and increased the hazard ratio (HR) to 2.1 (*p* value < 0.05) compared to those with low levels (Appendix A). We would like to prove our concept that blocking the interaction between S100A1 and RAGE with our new compounds could suppress colorectal cancer cell growth. To this end, we chose SW480 colorectal cancer cells, which express RAGE, as our experimental model in our study.

## 4. Conclusions

S100 proteins are released in paracrine–autocrine, endocrine, or both modes of secretion to influence cellular signaling in different cell types. Two-dimensional NMR ^1^H-^15^N HSQC experiments were used to find the interaction (binding) site of the S100A1 protein with pentamidine and the pentamidine derivative in a dimeric complex. The dissociation constant observed indicates that there was a fairly strong interaction between the protein and the ligand. Further, 2D NMR HSQC experiments strongly implied that S100A1-RAGE V domain complex binding occurred on its surface. The results indicate that the S100A1 protein binds to pentamidine and pentamidine derivatives via hydrophobic residues. We also calculated a model of S100A1-pentamidine and WLC-4059 heterodimer complexes using the HADDOCK program to locate the interaction site on the complex molecule. The structural complexes were superimposed, which clearly indicated that the pentamidine and WLC-4059 molecules blocked the interaction between S100A1 and the RAGE V domain. However, in the assay results, WLC-4059 showed more of a decrease in cell proliferation compared to pentamidine, and the cytotoxicity results showed that pentamidine is more toxic than the newly synthesized pentamidine derivative WLC-4059. Thus, these results illustrate that the pentamidine derivative WLC-4059 blocks the interaction between S100A1 and the RAGE V domain significantly by acting as an anti-proliferative agent that could be beneficial for therapeutic approaches for different types of cancers.

## Data Availability

The data is available in the manuscript and the Appendix A of this article.

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
