# Peer review of "The Anti-Cancer Activity of Pentamidine and Its Derivatives (WLC-4059) Is through Blocking the Interaction between S100A1 and RAGE V Domain"

_biomolecules, 2022, doi:10.3390/biom13010081_

Round 1
Author Response
Dear Reviewer,
We have attached the file for the response to the revision.
Best Regards
Nuzhat Parveen

Reviewer 2 Report
This manuscript entitled ”Inhibition of cell proliferation by pentamidine and its derivatives (WLC-4059) through blocking the interaction between S100A1 and RAGE V domain” is well-written. S100 proteins, a calcium-binding protein influence cellular signaling in different cell types. It is known that the RAGE V domain is one of S100A1 targets for binding to its hydrophobic surface, which in turn triggers signal-transduction cascades that activate cell growth, cell proliferation, and tumor formation. The authors used WLC-4059 (newly synthesized pentamidine analogs) to inhibit S100A1-RAGE V interaction.
The methodology of this manuscript is very complex. The authors applied (HADDOCK 2.2) software to generate the structural models of S100A1- pentamidine and the pentamidine derivatives.
The results of this study are clearly presented in the Figures.
Based results of this study, the authors concluded that the pentamidine derivative, WLC-4059 blocks the interaction between S100A1 and the RAGE V domain significantly by inhibiting cell proliferation which could be beneficial for therapeutic approaches for different types of cancers.
Comment:
Use the same font in the Conclusion
Author Response

(The authors gave the same response as above.)

Reviewer 3 Report
The authors of the manuscript "Inhibition of cell proliferation by pentamidine and its derivatives (WLC-4059) through blocking the interaction between S100A1 and RAGE V domain" in the introduction exhaustively describe both the two proteins (object of their study) and the chemical compounds used. The methods, results and discussion are equally clear and comprehensive. The bibliography is complete.
The topic treated by the authors seems to me interesting and partly carried out with a "new" vision, namely that of identifying compounds with antitumor activity that inhibit the protein-protein interaction, since this type of approach leads to an increase in the number of compounds with potential antitumor activity. Since the authors study pentamidine and its derivatives (WLC-4059) as potential compounds with antitumor activity, in my opinion, biological assays should be performed on multiple cell lines representative of the most common and / or more chemoresistant tumors (lung, pancreas, etc. .). They should also indicate in the title the anticancer activity of their compounds.
Author Response

(The authors gave the same response as above.)

Round 2
Reviewer 3 Report
I thank the authors for considering my suggestions.
Author Response
Dear Reviewer,
we have revised the manuscript as suggested.
thanks and best regards
Nuzhat.